# Low Serum Magnesium is Associated with Incident Dementia in the ARIC-NCS Cohort

**DOI:** 10.3390/nu12103074

**Published:** 2020-10-09

**Authors:** Aniqa B. Alam, Pamela L. Lutsey, Rebecca F. Gottesman, Adrienne Tin, Alvaro Alonso

**Affiliations:** 1Department of Epidemiology, Emory University School of Public Health, Atlanta, GA 30322, USA; alvaro.alonso@emory.edu; 2Division of Epidemiology and Community Health, University of Minnesota School of Public Health, Minneapolis, MN 55454, USA; lutsey@umn.edu; 3Department of Neurology, Johns Hopkins University School of Medicine, Baltimore, MD 21205, USA; rgottesm@jhmi.edu; 4Department of Medicine, University of Mississippi Medical Center, Jackson, MS 39216, USA; atin@umc.edu

**Keywords:** dementia, cognitive decline, magnesium

## Abstract

Higher serum magnesium is associated with lower risk of multiple morbidities, including diabetes, stroke, and atrial fibrillation, but its potential neuroprotective properties have also been gaining traction in cognitive function and decline research. We studied 12,040 participants presumed free of dementia in the Atherosclerosis Risk in Communities (ARIC) study. Serum magnesium was measured in fasting blood samples collected in 1990–1992. Dementia status was ascertained through cognitive examinations in 2011–2013, 2016–2017, and 2018–2019, along with informant interviews and indicators of dementia-related hospitalization events and death. Participants’ cognitive functioning capabilities were assessed up to five times between 1990–1992 and 2018–2019. The cognitive function of participants who did not attend follow-up study visits was imputed to account for attrition. We identified 2519 cases of dementia over a median follow-up period of 24.2 years. The lowest quintile of serum magnesium was associated with a 24% higher rate of incident dementia compared to those in the highest quintile of magnesium (HR, 1.24; 95% CI, 1.07, 1.44). No relationship was found between serum magnesium and cognitive decline in any cognitive domain. Low midlife serum magnesium is associated with increased risk of incident dementia, but does not appear to impact rates of cognitive decline.

## 1. Introduction

Minerals and their role in cognition have been attracting attention in dementia and cognitive decline research. Magnesium, in particular, has a potential beneficial effect on multiple morbidities, including diabetes, stroke, atrial fibrillation, and other cardiovascular diseases [1,2,3,4]. Given the role of these conditions as risk factors for cognitive decline, interest in the role of magnesium as a preventive or therapeutic approach in cognitive decline and dementia is growing. Magnesium’s potentially protective effect against cognitive decline has been demonstrated in several animal models [5,6,7], but human studies are limited with conflicting results, and mainly focus on dietary intake of magnesium [8,9]. The only study to date to examine the relation between serum magnesium and cognition within a cohort found a U-shaped—rather than linear—association of baseline serum magnesium with cognitive decline over a 10-year period (i.e.: high and low baseline serum magnesium were associated with increased risk of dementia) [10]. 

With the goal of growing our understanding on the role that magnesium can play in cognitive decline and dementia, we studied the association of mid-life circulating magnesium with incident dementia and cognitive decline over a 27-year period within a large community cohort.

## 2. Materials and Methods 

### 2.1. Study Population

The Atherosclerosis Risk in Communities (ARIC) study is an ongoing, community-based cohort study based in four communities across the US: Jackson, Mississippi; Washington County, Maryland; Forsyth County, North Carolina; and select suburbs in Minneapolis, Minnesota [11]. To date, participants have been examined in 7 visits spanning a 30-year time period, with the first visits taking place in 1987-1989. Participants have also taken part in regular phone calls (annual until 2012, twice-yearly thereafter). The recruited sample was exclusively black in Jackson, and representative of the underlying population in the other three sites (white and black in Forsyth County, and predominately white in Washington County and Minneapolis). For the purposes of this study, visit 2 (1990-1992) was considered the baseline, since it was the first time that cognitive function was assessed. This study has been approved by each study center’s institutional review board, with all ARIC participants having provided written informed consent at each visit.

Exclusion criteria for this analysis were based on the following: refusing consent for genetic testing for *APOE* genotyping (*n* = 45), prevalent dementia at visit 2 (*n* = 9), and missing magnesium measurements (*n* = 58). Additionally, because ARIC participants are predominately black and white, Asian and Native American participants were excluded (*n* = 40). Furthermore, black participants from Minneapolis and Washington County were excluded due to low counts (*n* = 50). Finally, those with missing covariates at visit 2 were also excluded (*n* = 2106). Selection into a secondary analysis looking at visit 5 as baseline can be found in Appendix A. 

### 2.2. Incident Dementia

ARIC utilized several approaches to ascertain dementia status. Briefly, starting from visit 5 (2011–2013), all participants who attended in-person evaluations underwent cognitive exams, with results of these assessments, along with earlier cognitive assessments (see below), evaluated and adjudicated by an expert committee. Of those that were unable or refused to attend visits, or to detect earlier cases of incident dementia (prior to visit 5), dementia status was based on over-the-phone dementia screeners and/or informant interviews. Additionally, hospitalization for, or death due to, dementia—as determined by ICD-9/10-CM discharge codes and/or death certificates—was classified as having dementia, with discharge codes reviewed from over the entire study follow-up. The methodology for diagnosing dementia in the ARIC study has been described in detail elsewhere [11]. 

### 2.3. Cognitive Function

We evaluated cognitive functioning at ARIC visits 2 (1990–1992), 4 (1996–1998), 5 (2011–2013), 6 (2016–2017), and 7 (2018–2019), using three cognitive tests designed to evaluate functioning in three cognitive domains. The delayed word recall test (DWRT) assesses verbal learning and short-term memory, by asking participants to memorize 10 words, use them in a sentence, and then recall the words again after a 5-minute break. The digit symbol substitution test (DSST) evaluates executive functioning by giving the participant 90 s to draw symbols that correspond to a set of numbers based on a key. The word fluency test (WFT) measures expressive language by having the participant list as many words as they can in 60 s that start with the letters F, A, and S. The testing procedures have been described in detail elsewhere [11]. 

We calculated z-scores for each test at all study visits, and then scaled the results to the visit 2 mean and standard deviation. We also generated z-scores for overall cognition by averaging the z-scores and standardizing to the visit 2 mean and standard deviation.

### 2.4. Serum Magnesium

Blood was drawn at visit 2 into vacuum tubes designated for either lipids or chemistries, and then centrifuged for 10 min at 3000× *g* at 4 °C. The blood samples were stored at −70 °C and then shipped to the ARIC central laboratories for analysis. Serum magnesium was measured using the metallochromic dye calmagite [1-(1-hydroxy-4-methyl-2-phenylazo)-2-napthol-4sulfonic acid] under the Gindler and Heth method [12]. Repeated measurements in 40 participants were sent to the same lab, with at least one week in between samples, to assess between-person variability (magnesium reliability coefficient = 0.69) and in-person variability (coefficient of variation = 3.6%).

### 2.5. Covariates

Along with assessing the impact of magnesium categorized into quintiles, we also examined associations of magnesium as a continuous variable, expressed as per standard deviation decrease of serum magnesium. Except for education status and diet scores which were measured at visit 1 and visit 3, respectively, analyses using visit 2 as baseline used covariates measured at visit 2, whereas analyses using visit 5 as baseline used covariates measured at visit 5. Educational attainment was categorized into 3 levels: did not complete high school; completed high school or general education development (GED) or 1-3 years of vocational school; or at least some college. The following blood analytes were measured from fasting blood samples taken at respective visits: estimated glomerular filtration rate, C-reactive protein, potassium, sodium, calcium, total and HDL cholesterol. Smoking and drinking status were self-reported. Blood pressure estimates were the average of the 2nd and 3rd measurements taken over the course of 15 min. Participant diets were assessed through principal component analysis using food frequency questionnaires to analyze the consumption of 32 food groups, generating scores characterizing how much of their dietary patterns were considered “Western” (characterized by consumption of meats, processed foods, sweetened drinks, etc.) and “prudent” (characterized by consumption of whole grains, fresh fruits and vegetables, etc.) [13]. The prevalence of coronary heart disease was based on the presence of myocardial infarction (MI) from adjudicated visit 1 electrocardiogram data, history of MI, or previous coronary bypass. Stroke history was based on prevalent stroke at visit 1 or stroke hospitalization before or at visit 2. Diabetes status was based on serum glucose (fasting cutoff ≥126 mg/dL; non-fasting cutoff ≥200 mg/dL), self-reported physician diagnosis of diabetes, or use of diabetes medication. 

### 2.6. Statistical Analysis

The association between serum magnesium (in quintiles) measured at baseline and incident dementia was measured using Cox proportional hazard models. Model 1 was adjusted for age, the combination of race and center (Jackson black people, Forsyth black people, Forsyth white people, Minneapolis white people, Washington white people), sex, and education (did not complete high school; high school graduate or vocational school; or at least some college). Model 2 adjusted for model 1 covariates, along with smoking history (ever smoked/never smoked), drinking status (current drinker/not current drinker), waist-to-hip ratio, diet scores, estimated glomerular filtration rate (mL/min/1.73m^2^), C-reactive protein (mg/L), potassium (mmol/L), sodium (mmol/L), calcium (mmol/L), coronary heart disease (prevalent/not prevalent), history of stroke (yes/no), systolic blood pressure (mmHg), diastolic blood pressure (mmHg), antihypertensive medication use (antihypertensive diuretic, non-diuretic antihypertensive, no antihypertensive use), total cholesterol-to-HDL cholesterol ratio, diabetes status (diabetes/no diabetes), and *APOE* ɛ4 carrier status (allele/no allele). The proportional hazards assumption was checked by testing interactions with log-time.

In a secondary analysis, to determine if more proximal measurements of magnesium (i.e.: magnesium in late life) influenced rates of incident dementia, we estimated the association between visit 5 serum magnesium and incident dementia, using Cox proportional hazard models.

We used linear models to assess cognitive decline from visit 2 through visit 7, based on visit 2 serum magnesium quintiles and fit them with generalized estimating equations (GEE), with an unstructured correlation matrix to account for repeated testing. Time was modeled using two linear spline terms with knots at 6 years (corresponding to visit 4) and 21 years (corresponding to visit 5) to account for the pronounced decline that is typically expected in later life. Models also incorporated interaction terms with the time splines and covariates. We also assessed differences in baseline cognitive function at visit 2 across magnesium quintiles to better understand baseline cognition. Additionally, in order to evaluate potential floor effects, we conducted a sensitivity analysis, excluding the bottom 5% of test scores within each race at baseline.

Significant attrition was expected over the course of 27 years [14], so we utilized multiple imputation with chained equations (MICE) in order to impute missing values [15]. Twenty datasets with imputed values were calculated based on dementia diagnosis, surveillance information based on telephone screeners and informant interviews, and visit 2 covariates. All cognitive decline results presented have been imputed.

All analyses were conducted using SAS 9.4 (Cary, NC; SAS Institute Inc).

## 3. Results

Figure 1 presents a flowchart for selection into the study, using visit 2 as baseline. There were 14,348 participants at visit 2, and after applying the exclusion criteria, the final analytic cohort included 12,040 participants free of dementia and with available magnesium at visit 2 (mean age: 56.9 years (SD: 5.7), 56.3% female, 24.6% black). The median follow-up time was 24.2 years (25th, 75th percentiles: 17.3, 27.1 years).

Participants with lower baseline serum magnesium were more likely to be female, black, and have less education than those with higher magnesium (Table 1). Those with lower serum magnesium were also more likely to be diabetic and on antihypertensive medication.

### 3.1. Incident Dementia

A preliminary analysis found no evidence of a U-shaped association when using the 3rd quintile as the referent group. Therefore, subsequent analyses use the 5th quintile as the referent group. When minimally adjusting for sex, race and center, age, and education, participants in the lowest quintile had a 34% higher rate of dementia [HR: 1.34; 95%CI: 1.17, 1.54] (Table 2). Additional adjustment for diet, cardiovascular disease correlates, APOE4 carrier status, and other micronutrients resulted in a slight attenuation, but remained significant, with a 24% higher rate of incident dementia [HR: 1.24; 95%CI: 1.07, 1.44]. A one SD lower serum magnesium concentration [SD: ~0.009 mmol/L] was associated with a 7% higher rate of dementia when adjusted for model 2 covariates [95%CI: 2%–11%]. When using visit 5 as baseline, no significant associations or trends were found between magnesium and incident dementia (Appendix A).

Race- and sex-stratified models revealed similar patterns of association, with no real differences found between black people and white people [race interaction, *p* = 0.51] nor between men and women [sex interaction, *p* = 0.46] (Appendix A). 

### 3.2. Cognitive Decline

Low magnesium was associated with poorer performance at baseline in the DSST, WFT, and the global composite score (Appendix A). Table 3 presents rates of cognitive decline during the 27-year follow-up period by serum magnesium at visit 2. There was no clear association between serum magnesium and cognitive decline, with magnesium modeled in quintiles for the composite cognitive score or any of the individual cognitive tests, even after excluding the bottom 5% of test scores at baseline (Appendix A).

## 4. Discussion

Within a large, community-based cohort, we found low levels of mid-life serum magnesium to be associated with an elevated risk of incident dementia, with a 24% increased risk of dementia for participants in the bottom compared to the top magnesium quintile, even when adjusting for demographics, lifestyle, cardiovascular risk factors, APOE4 carrier status, and other micronutrients. Mid-life magnesium, however, was not associated with decline over a 27-year period. No meaningful differences were found between race or sex. 

The discrepancy between incident dementia and decline may in part be explained by differences in baseline cognition and education levels. Cognitive performance at visit 2 was poorer among participants with lower serum magnesium compared to those with higher magnesium. Low magnesium participants also on average had less formal education than their higher magnesium counterparts, which has been shown to primarily affect baseline cognition while remaining unrelated to cognitive change [16]. Even after excluding the lowest 5% of scores at baseline to account for possible floor effects, rates of decline did not appear to differ across magnesium levels. Based on our findings, though the rates of decline do not seem to differ across magnesium levels, it may not take much decline to reach the dementia “threshold” for low magnesium individuals. 

Magnesium may target multiple pathways in dementia pathology. N-methyl-D-aspartate (NMDA) receptors play critical roles in learning processes and the formation of memories [17]. Through the glutamatergic excitation of NMDA receptors, calcium ions flow into cells and trigger other signaling pathways important in dementia and cognitive decline pathology [18]. Over-excitation of NDMA receptors, however, may impair synaptic activity and lead to neuronal necrosis [19]. Magnesium is able to block this NMDA-induced excitotoxicity by inhibiting NMDA receptors and subsequent cellular cascades [20,21]. Another pathway targets neuroinflammation triggered by beta-amyloid (Aβ) leakages through the blood brain barrier, which can lead to the release of proinflammatory cytokines, such as interleukins, tumor necrosis factor alpha (TNF-α), and nitric oxide [22,23], all resulting in a higher rate of neurodegeneration. Magnesium has shown to inhibit excessive Aβ production and prevent this inflammatory cascade [24]. Additionally, elevated magnesium may be able to induce amyloid precursor protein cleavage, which would also prevent the accumulation of Aβ [25].

Our results confirm much of the previous literature. Cross-sectional studies comparing those with diagnosed dementia against healthy controls found patients with dementia to have lower serum magnesium than their non-impaired peers [26]. In the Rotterdam study, serum magnesium at either extreme was associated with increased risk of dementia compared to the average serum levels (third quintile) [10]. Magnesium intake, whether through diet or supplements, seems to also be associated with better cognitive functioning in both mice and human models [7,27]. For instance, over a ten-year follow-up period, patients in Taiwan using magnesium oxide had a decreased risk of developing dementia compared to those not on magnesium oxide therapy [9]. Magnesium oxide is commonly prescribed as an antacid or laxative, and has been shown to increase serum magnesium [28]. Other studies have found benefits in having a more balanced intake of magnesium. In the Women’s Health Initiative Memory Study, women in the third quintile of dietary magnesium intake (corresponding to about 216–263 mg/day) had a lower risk of mild cognitive impairment or dementia than those in the first quintile; no association was found comparing the fifth quintile against the first quintile [8]. 

Though the evidence is promising, determining the utility of serum measurements versus dietary intake of magnesium in dementia pathology requires a critical examination of each of their shortcomings. Magnesium homeostasis is dependent on its interaction with calcium and phosphorus within the small intestines, bones, and kidneys [29]. Consequently, it is difficult to separate the effects of dietary magnesium from the overall quality of one’s diet. Conversely, though serum magnesium offers a more direct measurement of circulating magnesium than perhaps through supplement or dietary intake, serum magnesium has very little correlation with total body magnesium, most of which is present in either the bones and teeth or in the intracellular space [30]. When blood concentrations are low, magnesium may be pulled out of the cells to maintain normal levels in the blood [30], which could mask possible hypomagnesemia when determined through serum measurements alone. That said, though the association of blood magnesium with brain magnesium has been difficult to establish up to this point, previous research has found that both brain [31] and serum magnesium levels [32] are diminished in Alzheimer’s patients, suggesting some level of positive correlation between the two measurements.

Strengths of this study include the ability to use data from a large, prospective cohort study, which grants us significant time to follow-up with participants. We were also able to adjust our models for a variety of cardiovascular risk factors, diet, and other micronutrients. On the other hand, the limitations of this study warrant a cautious interpretation of the results. First, even though we were able to assess the impact of both mid-life and late-life magnesium on dementia and cognitive function, two measurements may not be enough to account for the impact of micronutrient fluctuations over a life course. Second, the cognitive change analyses may be influenced by possible “floor effects”, though we attempted to account for this in a sensitivity analysis by excluding the bottom 5% of scores within each race. Furthermore, though the MICE method for imputing missing data has been previously validated in the ARIC cohort [33], estimation errors may occur for data over 50% missing [34], and with over 80% of the cohort missing by visit 7, results should be interpreted with caution. Additionally, though we were able to follow-up with the cohort for over two and a half decades, because the baseline for this analysis corresponds to mid-life for our participants, we are unlikely to fully capture cognitive trajectories over a life course. Finally, though we adjusted for many biological and lifestyle factors and undertook the rigorous adjudication of dementia status, there may be some level of confounding by socioeconomic factors or possible malnutrition, biasing our results and potentially inflating the effect of serum magnesium in dementia pathology. 

## 5. Conclusions

In conclusion, consistent with some prior studies, we found that low circulating magnesium in midlife was associated with increased risk of dementia. Confirming this association across multiple forms of magnesium intake and measurements (i.e.: blood magnesium, brain magnesium) and elucidating the underlying pathways may offer new avenues for the prevention of dementia in the community.

## Figures and Tables

**Figure 1 nutrients-12-03074-f001:**
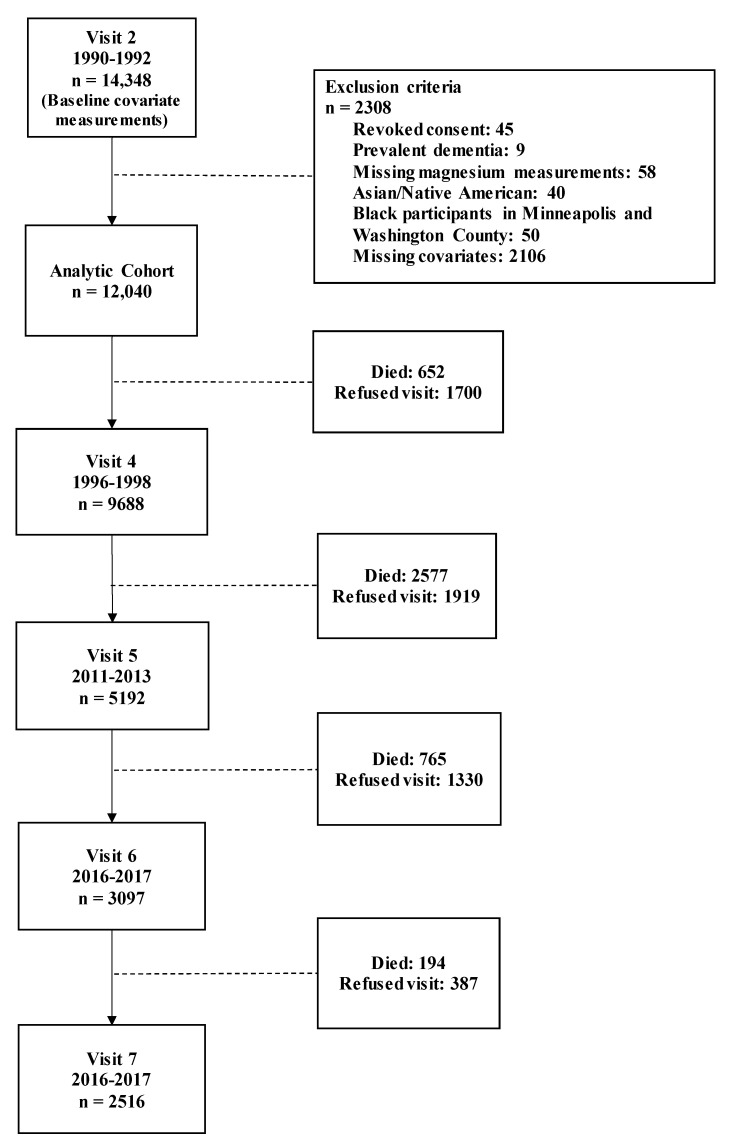
Diagram of the analytic cohort selection from visits 2 through 7 within Atherosclerosis Risk in Communities (ARIC) cohort.

**Table 1 nutrients-12-03074-t001:** Cohort characteristics by baseline magnesium quintile, ARIC 1990–1992.

Characteristics	Magnesium Quintiles				
	1st (≤1.4 mg/dL)	2nd (1.5 mg/dL)	3rd (1.6 mg/dL)	4th (1.7 mg/dL)	5th (≥1.8 mg/dL)
	N = 1650	N = 2370	N = 3255	N = 2599	N = 2166
Age, years	57.1 (5.8)	56.7 (5.8)	56.8 (5.6)	57.0 (5.7)	57.2 (5.7)
Female, %	63.0	56.6	56.7	54.0	53.0
African American, %	43.8	29.2	22.0	18.5	16.3
Education, %					
Did not complete high school	29.8	22.4	20.0	18.5	18.5
High school graduate and/or vocational school	39.1	41.1	42.3	42.2	42.6
At least some college	31.1	36.5	37.7	39.4	39.0
Sodium, mmol/L	140.3 (2.6)	140.6 (2.3)	140.8 (2.3)	141.0 (2.2)	141.2 (2.2)
Potassium, mmol/L	4.06 (0.42)	4.15 (0.38)	4.18 (0.38)	4.22 (0.40)	4.24 (0.40)
Calcium, mmol/L	0.521 (0.027)	0.518 (0.023)	0.518 (0.023)	0.519 (0.023)	0.520 (0.023)
Waist-to-Hip Ratio	0.93 (0.08)	0.93 (0.08)	0.92 (0.08)	0.92 (0.08)	0.93 (0.08)
Ever smoked, %	59.0	60.5	59.1	59.2	61.6
Current drinker, %	48.1	54.8	57.8	59.0	60.5
Prevalent coronary heart disease, %	6.8	6.4	5.9	5.2	5.2
Previous stroke, %	2.9	1.9	1.5	1.6	1.6
Diabetes, %	33.4	19.4	13.1	11.0	9.0
Systolic BP, mmHg	125.4 (20.4)	121.7 (19.0)	121.0 (18.2)	119.5 (17.9)	120.6 (17.8)
Diastolic BP, mmHg	73.1 (10.6)	72.2 (10.2)	72.1 (10.3)	71.6 (10.1)	71.9 (10.0)
Total Cholesterol-to-HDL cholesterol ratio	4.76 (2.06)	4.73 (2.00)	4.62 (1.81)	4.65 (1.74)	4.76 (1.93)
Antihypertensive medication, %					
Diuretic	25.5	15.7	12.3	10.7	9.4
Non-diuretic, antihypertensive	5.0	3.0	3.1	3.3	2.6
No antihypertensive medication	69.6	81.3	84.7	86.0	88.0
eGFR, mL/min/1.73m^2^	97.2 (20.1)	96.1 (17.2)	96.1 (16.1)	94.5 (15.7)	92.3 (16.3)
*APOE* ε4 allele, %	31.4	31.2	30.8	29.6	30.9
C-reactive Protein, mg/L	5.9 (8.8)	4.6 (6.9)	4.2 (7.1)	3.9 (7.1)	3.8 (6.3)
Western Diet Score	−0.018 (0.981)	−0.012 (1.000)	−0.031 (0.979)	−0.046 (0.967)	−0.009 (0.996)
Prudent Diet Score	−0.025 (0.988)	0.021 (1.027)	0.011 (0.950)	0.032 (0.987)	0.019 (1.00)

Values correspond to mean (standard deviation) or percentage.

**Table 2 nutrients-12-03074-t002:** Association of baseline magnesium with incident dementia, ARIC 1990-2019.

Quintile of Magnesium	Person Years of Follow-Up	Dementia Cases	IR ‡	Model 1 * HR (95%CI)	Model 2 ** HR (95%CI)
Quintile 1	32,306	367	11.36	1.34 (1.17, 1.54)	1.24 (1.07, 1.44)
Quintile 2	49,507	475	9.59	1.11 (0.97, 1.26)	1.08 (0.95, 1.24)
Quintile 3	70,753	681	9.63	1.03 (0.91, 1.16)	1.03 (0.91, 1.16)
Quintile 4	56,470	559	9.90	1.07 (0.95, 1.21)	1.08 (0.95, 1.22)
Quintile 5	47,188	437	9.26	1 (Ref)	1 (Ref)
Per 1 standard deviation (~0.009 mmol/L) decrease in Mg				1.09 (1.05, 1.14)	1.07 (1.02, 1.11)

HR, hazard ratio; 95%CI, 95% confidence interval. ‡ Crude incidence rate, per 1000 person-years. * Model 1 adjusted for sex, race and center, education, and age. ** Adjusted for Model 1 variables, along with history of smoking, drinking status, western and prudent diet scores, waist-to-hip ratio, estimated glomerular filtration rate, c-reactive protein, sodium, potassium, calcium, prevalent coronary heart disease, previous stroke, antihypertensive medication use, systolic and diastolic blood pressure, total-cholesterol-to-HDL cholesterol ratio, apolipoprotein E4 carrier status, diabetes status.

**Table 3 nutrients-12-03074-t003:** Cognitive change over 27 years by baseline magnesium quintile, ARIC 1990-2019.

Test	Quintiles	Model 1 *	Model 2 **
Global			
	1	−0.031 (−0.092, 0.029)	0.003 (−0.060, 0.067)
	2	0.052 (−0.0004, 0.104)	0.063 (0.011, 0.115)
	3	0.027 (−0.022, 0.075)	0.030 (−0.017, 0.077)
	4	−0.021 (−0.074, 0.031)	−0.022 (−0.073, 0.028)
	5	0 (Referent)	0 (Referent)
Per 1-SD *** decrease in Mg		0.004 (-0.013, 0.020)	−0.003 (−0.022, 0.016)
DWR			
	1	−0.023 (−0.126, 0.080)	0.001 (−0.106, 0.107)
	2	0.058 (−0.033, 0.149)	0.066 (−0.027, 0.158)
	3	0.052 (−0.035, 0.139)	0.053 (−0.034, 0.141)
	4	−0.051 (−0.145, 0.043)	−0.054 (−0.146, 0.038)
	5	0 (Referent)	0 (Referent)
Per 1-SD decrease in Mg		0.011 (−0.017, 0.039)	0.007 (−0.028, 0.042)
DSS			
	1	−0.039 (−0.096, 0.017)	−0.0004 (−0.056, 0.055)
	2	0.048 (0.006, 0.090)	0.063 (0.021, 0.105)
	3	0.020 (−0.018, 0.059)	0.027 (−0.011, 0.065)
	4	−0.023 (−0.069, 0.022)	−0.023 (−0.068, 0.022)
	5	0 (Referent)	0 (Referent)
Per 1-SD decrease in Mg		0.002 (−0.014, 0.017)	−0.010 (−0.027, 0.008)
WF			
	1	−0.017 (−0.082, 0.047)	0.018 (−0.052, 0.089)
	2	0.034 (−0.017, 0.085)	0.047 (−0.003, 0.097)
	3	0.009 (−0.041, 0.059)	0.011 (−0.038, 0.061)
	4	0.013 (−0.039, 0.064)	0.012 (−0.039, 0.063)
	5	0 (Referent)	0 (Referent)
Per 1-SD decrease in Mg		−0.001 (−0.018, 0.017)	−0.007 (−0.030, 0.015)

SD, standard deviation; DWR, delayed word recall; DSS, digit symbol substitution; WF, word fluency. * Model 1 adjusted for sex, race and center, education, and age, and interactions of all covariates with time. Time modeled as spline term with knots at 6 years and 21 years. ** Adjusted for model 1 variables, along with history of smoking, drinking status, western and prudent diet scores, waist-to-hip ratio, estimated glomerular filtration rate, c-reactive protein, sodium, potassium, calcium, prevalent coronary heart disease, previous stroke, antihypertensive medication use, systolic and diastolic blood pressure, total-cholesterol-to-HDL cholesterol ratio, apolipoprotein E4 carrier status, and diabetes status, and interactions of all covariates with time. Time modeled as spline terms with knots at 6 years and 21 years. *** SD being equal to ~0.009 mmol/L.

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
