# Peer review of "Low Serum Magnesium is Associated with Incident Dementia in the ARIC-NCS Cohort"

_nutrients, 2020, doi:10.3390/nu12103074_

Round 1

Reviewer 1 Report

This is a highly interesting study. There are only 2 minor points to be omproved:

Table 1: Calcium concetrations should also be presented in mmol/L instead of mg/dL. - Similarly, Mg should also be presented in mmol/L (Lines 172, 177, 203).

Discussion/Line 257: Central nervous Mg effects (described in this study) should be correlated to brain-Mg (liquor, tissue-Mg etc.). The difficulties arising to correlate "brain-Mg" with "blood-Mg" are known but shold shortly be discussed. Perhaps it should be also proposed to study, e.g.,  liquor-Mg in further studies, whenever possible.

Author Response

  • Point 1: “Table 1: Calcium concentrations should also be presented in mmol/L instead of mg/dL. - Similarly, Mg should also be presented in mmol/L (Lines 172, 177, 203).”
    • Calcium concentrations in mg/dL were converted to mmol/L (line 172, Table 1).
    • Magnesium concentrations were also converted from mg/dL to mmol/L [lines 181, 185 (Table 2), 213].
  • Point 2: “Discussion/Line 257: Central nervous Mg effects (described in this study) should be correlated to brain-Mg (liquor, tissue-Mg etc.). The difficulties arising to correlate "brain-Mg" with "blood-Mg" are known but should shortly be discussed. Perhaps it should be also proposed to study, e.g., liquor-Mg in further studies, whenever possible.”
    • We appreciate the thoughtful addition to the discourse on magnesium measurements. We have added a line briefly addressing the relationship between brain and blood magnesium to the discussion section: “…though the association of blood magnesium with brain magnesium has been difficult to establish up to this point, previous research have found both brain [31] and serum magnesium levels [32] are diminished in Alzheimer’s patients, suggesting some level of positive correlation between the two measurements.” (lines 267-270) We have also briefly addressed the need for further research in the conclusion paragraph: “Confirming this association across multiple forms of magnesium intake and measurements (i.e.: blood magnesium, brain magnesium) and elucidating the underlying pathways may offer new avenues for the prevention of dementia in the community.” (lines 291-292)

Reviewer 2 Report

The authors report results on the association between serum magenesium levels and dementia in a large sample studied over more than 20 years.

The research is plausible and the results are critically discussed pointing out also the limitation of the study such as counfounding influences.

The following revisions are necessary:

Line 22: d before The?

Throughout: the in-text references do not completely correspond to authors‘ guidelines (no space between numbers, fullstop after and not before reference number)

Line 160: the sentence appears incomplete, verb missing

Author Response

  • "The authors report results on the association between serum magenesium levels and dementia in a large sample studied over more than 20 years.

    The research is plausible and the results are critically discussed pointing out also the limitation of the study such as counfounding influences."

    • We appreciate your positive review.
  • Point 1: “Line 22: d before The?” and “Line 160: the sentence appears incomplete, verb missing”
    • Thank you for catching these typos. Corrections have been made in lines 23 and 169.
  • Point 2: “Throughout: the in-text references do not completely correspond to authors’ guidelines (no space between numbers, fullstop after and not before reference number)”
    • The reference formatting issues have been corrected as suggested throughout the paper (not tracked).